# Instantaneous LEM back-analyses of major rockslides triggered during the 2016-2017 Central Italy seismic sequence

Luca Verrucci[1], Giovanni Forte[2], Melania De Falco[2], Paolo Tommasi[3], Giuseppe Lanzo[1], Kevin W. Franke[4], Antonio Santo[2]

[1] Dipartimento d'Ingegneria Strutturale e Geotecnica, Sapienza Università di Roma, Rome, Italy
[2] Dipartimento d'Ingegneria Civile, Edile e Ambientale; Università degli Studi di Napoli Federico II, Naples, Italy
[3] Cnr – Istituto di Geologia Ambientale e Geoingegneria, Roma, Italy
[4] Department of Civil and Environmental Engineering, Brigham Young University, Provo, UT, USA

*Correspondence to*: Giovanni Forte (giovanni.forte@unina.it)

**Abstract.** Among the almost 1400 landslides triggered by the shocks of the 2016-2017 Central Italy seismic sequence, only a limited number, all classifiable as rockslides, involved volumes larger than 100 m³. Four of these failures, including the three largest among the documented landslides, were described in terms of structural and geomechanical investigations in a previous paper. In this study, the estimated acceleration time histories at the rockslide sites were evaluated through a 2D simplified numerical model accounting for the attenuation phenomena and for the topographic effect of the rock cliffs from which the slide detached. Instantaneous stability analyses were carried out to obtain insights into the variability of the instantaneous margin of safety along the motion, over the entire spectrum of mechanisms that could be activated. Finally, some general suggestions on the pseudo-static verification method for three-dimensional cases are proposed, which represent useful indications to hazard evaluation at local and regional scale.

## 1 Introduction

Most of the landslides occurred in Central Italy during the earthquakes of the last century were rockfalls and rockslides (**Martino *et al.* 2017; Esposito *et al.* 2000**). Rock falls were few cubic meters in volume whilst rockslides involved volumes lower than 40000 m³. Even though large rockslides have been recorded during moderate-to-large magnitude seismic events (see e.g. **Sepulveda *et al.* 2016**), type and size of rock slope failures occurred during Central Italy earthquakes is compatible with their level of released energy (moment magnitude, $M_w \leq 6.5$) and to the lithology of the formations outcropping in the areas close to the seismogenic structures. They involved relatively small rock volumes in comparison to the relationships reported in literature (**Malamud *et al.* 2004**, **Marc *et al.* 2017**), although similar rock slope failures were recorded in the 1976 Friuli seismic sequence (**Govi and Sorzana, 1977**), which was characterized by comparable seismic input and surface geology. Similarly to many other earthquake-triggered landslides (**Rodriguez *et al.* 1999**), the landslides occurred during the Central Italy earthquakes were characterized by a marked disruption of the rock mass and originated on very steep slopes,

where inertial forces easily remove well-delimited and scarcely constrained blocks from the slope through rigid sliding/toppling or tensile failures of overhanging blocks. All these instability phenomena are very "brittle", i.e., relatively small displacements develop before the constraints change abruptly and a fast propagation phase begins with a free fall of single or multiple non-interacting blocks (**Esposito *et al.* 2000**, **Lanzo *et al.* 2009**, **Stewart *et al.*, 2018**; **Franke *et al.* 2019**).

Since seismic loading only acts at the early detachment of earthquake-triggered rock failures (propagation is controlled only by gravity loading and slope geometry) the study of this stage is very important for hazard evaluation. Literature has evidenced that the intrinsic key factors influencing the seismic activation of these landslides include the structural features of the rock mass (e.g. joint spacing and orientation, presence of major joints), and the hydraulic conditions (e.g., **Massey *et al.*, 2017; Tsou *et al.*, 2018**). Also the topographic modifications to ground motion and the effects of step-like slope topography
on seismic motion have been studied by many authors as **Ashford *et al.* (1997)**, **Bouckovalas & Papadimitriou (2005)**, **Nguyen & Gatmiri (2007)**, **Sepúlveda *et al.* (2005a,b)**, **Pagliaroli & Lanzo (2008)**, **Lenti & Martino (2012)**, **Li *et al.* (2019) and Pignalosa *et al.* (2022)**. **Li *et al.* (2019)** described parametric analyses of steep rock slopes, providing the amplification factors for only weak seismic excitations and on the assumptions of homogeneous and elastic rock materials. Nevertheless, the general influence of very steep and vertical slopes, like those from which the rockslides detached, is
scarcely investigated in detail in the literature and it is a valuable topic to investigate. Also the amplification distribution along the slope profile, that is important to estimate the inertial effects on the surface rock blocks, is not enough evidenced, and only quite rough approximations are applied, such as a linear variation between the crest and the toe (**Mavrouli *et al.*, 2009**).

The fundamental method used to study and estimate the block stability in a 3-dimensional approach is the limit
equilibrium method for rock wedges (**Wittke, 1967**; **Goodman, 1976**; **Hoek & Bray, 1977**). The method was extended to the dynamic conditions by **Ling & Cheng (1997)** through a pseudo-static approach that was also experimental verified by **Kumsar *et al.* (1997)**. However in these works the 3-dimensionality of the seismic action is not usually taken into account as the inertial forces are applied in the direction normal to the slope. This practice can disguise the possible activation of different mechanisms during shaking because the real mechanism of initial failure is determined by the direction of the
resultant external force on the block (**Goodman, 1976**).

These considerations sparked investigation on the relationship of the rock mass structure (especially the orientation of major discontinuities) with the failure mechanism and, in particular with the evolution during the seismic shaking, for some of the largest rockslides among those reported during the 2016-2017 Central Italy Seismic Sequence (CISS). Analyses were conducted with a 3D instantaneous limit equilibrium method (LEM) by applying the acceleration histories of the main
earthquake shocks modified by a simple local viscoelastic seismic response for topographic conditions. Such an insight, when transferred to predictive stability analyses, can lead to a better awareness of the possible mechanisms and hence to a more effective evaluation of the hazard and of the successive fall/avalanche stage, which still represent a challenging problem (Wartman et al. 2019).

## 2 Examined rockslides and available data

The CISS consisted, from August 2016 to mid-January 2017, of several shocks ranging from $M_w$ 5.0 to $M_w$ 6.5 (the latter is the highest magnitude recorded in Central Italy during the last century, **Rovida et al., 2019**) and involving nearly 1500 km$^2$ of the regional normal fault system affecting an area characterized by a seismic gap between the 1997 Mw 6.1 Colfiorito-Sellano earthquakes to the north and the 2009 Mw 6.1 L'Aquila earthquake to the south). The seismic events caused more than three hundred casualties, heavily damaging the physical environment, buildings and historical heritage as well (**Miano et al., 2020**; **Saretta et al., 2021**) and triggered more than 1370 landslides, mainly rock falls and slides, affecting limestone formations and to a lesser extent, the flysch units. The shocks with $M_w$ >5 triggered the largest rockslides, which mobilized volumes up to 35000 m$^3$. Their features were described by **Forte et al. (2021)**, **Lanzo et al. (2019), Franke et al. (2019) and Romeo et al. (2017)**. After the seismic sequence, input data on the rockslides were collected during several investigation campaigns that included aero-photogrammetric surveys with unmanned aerial vehicles (UAVs), sampling of blocks and joints and direct *in-situ* measurements of joint orientation, spacing and roughness. Data and results of the reconnaissance investigations are reported in detail by **Forte et al. (2021)**, who described the local geology, rock mass structure, major joints delimiting the failed mass and the volumes.

Main features of the selected four rockslides and their possible triggering earthquakes are shown in **Table 1**, while their location is reported in **Figure 1** together with the epicentres of the main shocks of the CISS. **Table 2** summarizes the main features of the shocks that triggered the rockslides, obtained from signals recorded at the neighbouring accelerometric stations on stiff ground (Engineering Strong-Motion Database, ESM, **Luzi et al. 2016**).

These rockslides were chosen because they represent four of the largest failures among those detected during the reconnaissance field surveys conducted immediately after the seismic shocks (Costa Cattiva and Nera rockslides) or the most accessible among those observed on aerial images taken soon after the end of the seismic sequence (Piè la Rocca and Rubbiano rockslides). In this way, UAV surveys, which allowed detailed morphological and geo-structural setting, could be conducted in a relatively short time after the seismic sequence. Other large rockslides detected on aerial images, with much higher logistic issues, were successively investigated and are currently being analyzed in the framework of national research projects.

They originated from the limestones formations known as *Calcare Massiccio Fm.* and *Maiolica Fm.*, and were studied by merging classical field methods with newer remote sensing approaches by UAV. Comparison of geostructural analyses at the scale of the slope with the regional tectonic setting indicated that all the four rockslides are locally characterized by major discontinuities of the older anti-Apennines sets (NE-SW) despite a much higher frequency of the quaternary Apennines tectonic sets (NW-SE). The failures often occurred following the breakage of rock bridges during the seismic shaking, as pointed out by stability analyses and evidence on 3D models. **Figures 2a** through **2d** present post-collapse frontal views of the rockslides. The bedrock is made of limestone for all the four cases, which is either layered (Costa Cattiva, Nera and Rubbiano rockslides) or relatively massive (Piè la Rocca rockslide). The four rock slopes are all very steep and three of them

(Nera, Piè la Rocca and Rubbiano) are located within tectonically disturbed zones: (reverse fault and associated fold hinge, a fault zone, and a thrust front, respectively). The wedges were all delimited by near-planar single major joints (labelled in **Figure 2**), excepting for the Rubbiano rockslide, which was delimited at its back by a surface resulting from the combination of several discontinuities of limited extent. **Figure 2** also includes stereo-plots with great circles of the planes delimiting each wedge at the very beginning of the detachment, as estimated from the 3D models and point clouds obtained from UAV aerial surveys (**Franke *et al.* 2019; Tommasi *et al.* 2019**). Great circles refer to single major joints or to planes interpolating combinations of minor joints. Low-dip joints (i.e., along which shear occurred) showed negligible intact rock bridges excepting for that delimiting at the base the Nera rockslide and its contribution to shear strength was therefore considered. Portions of intact rock were found along the subvertical surfaces delimiting the back of two of the failed wedges, where they provided some tensile resistance (Piè la Rocca and the Rubbiano rockslides). The latter was large enough to deserve consideration in the stability analyses.

Observations of Google Earth satellite images taken at different dates over 2016, indicates that Nera and Rubbiano rockslides occurred during the strongest shock of the seismic sequence (Norcia, October 30[th], $M_w$= 6.5). Piè la Rocca slide occurred during one of the two August 24[th] shocks: Accumoli $M_w$= 6.0 or Norcia $M_w$= 5.3, the latter having an epicentre very close to the site (4.2 km). Costa Cattiva rockslide occurred during one of the two October 26[th] shocks: either Visso $M_w$= 5.4 or Castelsantangelo $M_w$= 5.9. Locations of the epicentres are reported in **Figure.1**.

Since the failed rock slopes were not accessible for geophysical investigation, the shear wave velocity, $V_s$, used in seismic response assessment was estimated based on results of borehole geophysics conducted on the same geological formations at neighbouring sites having similar fracturing and loosening of the rock mass. Down holes in the *Maiolica formation* conducted in the framework of seismic microzonation of the struck area (*Banca dati microzonazione sismica,* www.webms.it) indicated that pervasively fractured rock (i.e. with RQD values close to 0) exhibits a $V_s$ of about 600 m/s, which increases to 2000 m/s in a fairly jointed rock mass.

The strength parameters adopted for the stability analysis of the rockslides, which are the same used by **Forte *et al*. (2021),** were derived from direct in situ investigations at the rockslide sites and on laboratory tests on samples collected both at the rockslide sites and at neighbouring sites in the same geological formations. The friction angle along the sliding planes varies between 40° and 47° depending on the local roughness and waviness. Intact rock bridges along the joints forming the slide scar, which broke during failure, were observed on close UAV images. Their contribution to joint cohesion was evaluated as the cohesive component of the shear strength of the rock mass ($c_{rb}$ = 570 kPa) multiplied by the area $A_{rb}$ of the rock bridges. The parameter $c_{rb}$ was estimated by linearizing the Hoek Brown strength envelope of the rock mass, obtained from the strength envelope of the rock material scaled through the Geological Strength Index, *GSI*, determined on the rock outcrops at the sites (**Hoek *et al.* 2002**).

High resolution imagery captured from UAVs and during helicopter surveys over the Nera slide also revealed that the tip of the sliding surface appeared to be irregular and paler than the surrounding rock mass. This evidence induced **Forte *et al.* (2021)** to hypothesize that the lower part of the failure surface developed through the rock mass rather than along an

existing joint. Therefore, an additional contribution was considered by multiplying the cohesion $c_{rb}$ by the area (800 m$^2$) of the failure surface at the wedge tip.

For the Rubbiano rockslide (RB), a tensile strength equal to 10% of the rock mass cohesion $c_{rb}$, was considered as an additional strength contribution that contrasted the detachment from plane 1 (**Figure 2d**). Where the plane 3 is present (as at the top of Piè la Rocca rockslide, **Figure 2c**) the wedge detaches along it from the rock mass behind, thus providing no strength contribution. Geometry and strength parameters adopted in the static LEM back analyses described by **Forte *et al*. (2021)** are shown in **Table 3**.

## 3. Method of analysis

For each rockslide, after a kinematic analysis in static conditions, the seismic motion responsible of failure was estimated in two steps: a Ground Motion Prediction Equation (GMPE) was applied first to the available ground motion records to account for attenuation at the rockslide sites, then modifications to ground motion induced by the slope morphology were evaluated through a general simplified 2D numerical model that reproduces the main resonance and attenuation phenomena affecting very steep rock slopes during seismic shocks.

Since the investigated slopes can be roughly assimilated to steep flanks of deep valleys (200-500 m) separated by large and relatively flat mountain ridges, the modifications to the ground motion that the general slope morphology produces at each site were estimated through a finite difference model that simulates the visco-elastic dynamic behaviour of a simplified slope: a step-like slope with a vertical cliff of height $H$ and upstream and downstream horizontal areas (**Figure 3**).

In the frequency domain, the modification of a harmonic motion of wavelength $\lambda$ (frequency $f$) propagating in a medium with shear wave velocity $V_S$ and Poisson ratio $v$, can be expressed by the amplification ratio $A$ between the amplitude at a point at height $h$ on the slope face and the amplitude on the horizontal rigid outcrop. In a dimensional analysis approach, $A$ can be expressed as a function of the dimensionless variables $\zeta, \eta, v$, being $\zeta = h/H$ and $\eta = H/\lambda = Hf/V_S$.

If planar vertically propagating waves that are polarized are considered, two cases (P- and S-waves) are sufficient to estimate the variability of $A$ along the entire vertical wall ($\zeta$ in the range 0.0-1.0). To cover a sufficiently broad frequency range ($\eta$=0.03-2.0), two analyses were performed for each wave type using Ricker waves with different frequency content as input motion.

The model, built in the 2D finite difference code FLAC (**Itasca 2011**), consists of a 100 m high slope. The model bottom is an absorbing viscous boundary, whilst free-field boundary conditions are applied to the lateral boundaries, which are located at least five times $H$ from the slope face (**Figure 3**). A Rayleigh formulation was assumed with a uniform critical damping ratio of $D = 0.5\%$. Elastic properties $V_S = 100$ m/s and $v = 0.3$ were considered, but the normalized results can be extended to a cliff of different height and stiffness thanks to the principle of linear superposition.

Finally, the behaviour of the rock wedges during the shocks was analysed using the instantaneous limit equilibrium procedure in the instrumental hypothesis that they moved rigidly with the surrounding rock mass. The volumes of the

rockslides are in fact small enough to consider the primary mechanism of failure as a rigid wedge (**Hungr *et al.* 2014**) and to explore the role of the inertial forces in inducing failure. At this early stage, very small displacements occurred mainly as sliding, and the constraint configuration was dictated by the original orientation of pre-existing joints. The possible subsequent kinematic evolution (e.g. toppling), the disarrangement of the wedge and the start of the propagation phase are out of the scope of this paper. The estimated seismic input was applied to calculate the time histories of the safety factors during the shocks for the rockslides under the same assumptions of static LEM back analyses conducted by **Forte *et al.* (2021)**: the landslide body is subjected to gravity only, water pressure is absent, sliding surfaces are planar and a Mohr-Coulomb strength criterion is assumed. Only translational sliding mechanisms were considered, because their predominant role in driving the wedges to the collapse clearly emerged (**Forte *et al.*, 2021**).

Variable inertial forces $\vec{I}(t) = -m\vec{A}(t)$, applied uniformly to the rigid blocks with mass $m$, were added to equilibrium equations. This procedure is mechanically consistent only as long as the block does not displace with respect to the rock mass; therefore it only provides a realistic assessment immediately before sliding begins. In fact, the relative motion alters the inertial forces with respect to those calculated with the base acceleration and furthermore reduces the strength due to progressive smoothing of the joint surface and failure of the rock bridges. For these reasons the safety factor ($F_S$) during seismic excitation calculated through this analysis is intended to identify the most probable instants of failure initiation and the critical mechanisms. This type of calculation also represents an instrument to weight the relative importance of sliding mechanisms during shaking and thus to better handle the pseudo-static analysis method. Safety factors is calculated for the whole shaking duration and in turn it could assume also values lower than 1.0 during some time intervals.

In resolving equilibrium and calculating $F_S$, the activation of a different translational mechanism with respect to that occurring in static conditions was also considered. In fact, the instantaneous sliding mechanism is controlled by the current direction of the resultant external force, which in a dry slope coincides with the sum of the block weight and the inertial force. The number of passages between different mechanisms during the seismic event is related to the oscillating amplitude of the resultant force and to the distance of its pole from the kinematical region boundaries.

The instantaneous $F_S$ for the Mohr-Coulomb strength criterion can be calculated trough Eqs. (1) and (2) in case of sliding along a single plane or both planes, respectively:

$$F_S = \frac{cA + N\,tg\varphi}{T} \tag{1}$$

$$F_S = \frac{c_1A_1 + N_1tg\varphi_1 + c_2A_2 + N_2tg\varphi_2}{T_{12}} \tag{2}$$

$N$ and $T$ in Eq. (1) are the normal and tangential components of the resultant acting on a single sliding plane. $T_{12}$, $N_1$ and $N_2$ in Eq. (2) are the components of the resultant force parallel to the intersection line $I_{12}$ of planes 1 and 2 and normal to the planes, respectively. $c$, $\varphi$ and $A$ are the cohesion, friction angle and the contact areas, respectively; subscripts refer to the plane. The passage from one mechanism to another one entails an instantaneous change in the $F_S$ value.

## 3 Seismic input at the rockslide sites

The estimate of peak ground acceleration, PGA, calibrated with an appropriate GMPE at each of the four rockslide sites is presented in **Figure 4**. Diamonds represent the measured PGAs of the horizontal components (i.e., geometric mean of East and North components) versus the Joyner–Boore distance ($D_{JB}$) of the station. These measurements were interpolated with and calibrated against the GMPE of **Bindi** *et al.* **(2011)** (blue solid lines in **Figure 4**). The site class A (rigid ground, according to Eurocode 8, **EN 1998-5:2004**). and the normal fault class were used in the GMPE, while the moment magnitude was used as the regression parameter. The PGA at the sites were finally estimated using the $D_{JB}$ of each site on the interpolated GMPE (empty blue circle symbols in **Figure 4**). The input accelerograms were obtained by linearly scaling the recordings at the closest station on rock outcrop to the estimated PGA; the same scaling factors, $S$, was used for all the components. The parameters used in both the GMPE calibration and the scaling procedure are shown in **Table 4**.

For the estimate of the amplifications effects at the four sites the plots derived from the finite difference model described in section 2 were utilized (**Figure 5**). The plots report amplification ratios of the normal ($A_n^P$, $A_n^S$) and vertical ($A_z^P$, $A_z^S$) component, for the incoming P- and S-waves, and are functions of $\zeta$ and $\eta$. Along a horizontal line (i.e., at constant $\zeta_0$), the diagrams give the amplitude of the transfer functions from the outcrop motion (horizontal and vertical component, respectively, for the incoming S- and P-waves) to the motion of a point at height $h=\zeta_0 H$ on the cliff.

According to literature results (**Ashford** *et al.* **1997; Assimaki** *et al.* **2005**), for the incident S waves the most amplified wavelength corresponds to the first normalized modal frequency of 0.2, with a peak of $A_n^S$ greater than 1.4. For the Nera rockslide ($H$= 400m), the main resonance frequency, which is about 1.0 Hz, is critical along almost the entire cliff, although amplification decreases as elevation decreases. In addition, at medium and lower elevations, the higher frequencies are reduced overall. The vertical component produced by incident S-waves has significant amplitude ratios $A_z^S$ only at the crest and for normalized frequencies in the wide range 0.4-1.4 (e.g., 2.0 – 7.0 Hz for the Nera case with about $V_S$ = 2 km/s).

The amplification ratios for incident P waves, $A_n^P$ and $A_z^P$ reveal a main amplification of the vertical component at the crest and at almost the whole vertical wall for a normalized frequency of about 0.1 (0.5 Hz for the Nera case). Conversely, the horizontal component is flattened all along the cliff wall for all frequencies ($A_n^P$ < 0.8).

The linear process used to assess the motion at the elevation of the centre of gravity of the rockslide proceed as follows. Since the normal (horizontal) and vertical components $a_n$ and $a_z$ of the outcrop acceleration can be considered equivalent, respectively, to an S- and a P-wave in a vertical plane normal to the slope face, their Fourier transform $a_n(\eta)$ and $a_z(\eta)$ are multiplied to the transfer (amplification) functions to obtain the output components on the cliff (i.e., after morphological modifications). These are successively combined:

$$a_{out,n}(\eta) = A_n^S(\eta,\zeta_0)a_n(\eta) + A_n^P(\eta,\zeta_0)a_z(\eta) \tag{3a}$$

$$a_{out,z}(\eta) = A_z^S(\eta,\zeta_0)a_n(\eta) + A_z^P(\eta,\zeta_0)a_z(\eta), \tag{3b}$$

where $\zeta_0 = h_0 / H$ is the normalized height of the rockslide gravity centre. Finally the acceleration vector $\vec{A}(\zeta_0,t)$ is obtained by applying the inverse Fourier transform to the (3) and assuming a shear wave velocity of 2200 m/s at all sites. The hypothesis

that the seismic response develops in plane strain condition can be assumed for a very long cliff or valley and therefore the component $a_p(t)$ can be considered unmodified. The geometrical and morphological features that control the response calculations are shown in **Table 5** for each rockslide.

In **Figure 6** the amplification/attenuation effects are described by comparing the acceleration response spectra (damping 5%) of the two motions for all the considered shocks and for both the horizontal (normal) and vertical components. The modification of the motion is usually significant for periods lower than 1-2 s, while it is negligible for periods higher than the fundamental period $T_0$ of the first vibration mode of the cliff ($\lambda = 5H$) indicated by vertical dotted lines.

These analyses aim to estimate the general modifications to the seismic motion caused by large scale morphological features. Nonetheless, local irregularity of the slope surface like sharp ridges, spurs and pinnacles, which can induce significant further local amplifications/attenuations especially for small volumes, are not considered in the present research.

## 4 Seismic back analysis

The visual conception of the possible mechanism switches calculated during the instantaneous LEM analysis is represented by the kinematical regions reported in **Figure 7** for each rockslide**.** The regions are spherical triangles identified by the directions of the normal vectors to the planes and the directions of the plane intersection lines (**Londe *et al.* 1969**). Therefore the calculated (instantaneous) direction of the resultant external force on the block defines a different sliding mechanism depending on which triangle it belongs to. The directions of the initial static resultants (block weight) are indicated through red circles but during the seismic analyses they move around and can cross over the triangle boundaries.

For each analysis, the instantaneous activated mechanisms and the time histories of $F_S$ are reported in **Figure 8**. For instance, the mechanism of the Nera rockslide (**Figure 8a**) changes from sliding along the $i_{12}$ to sliding along the single plane 2. The latter mechanism has a quite lower level of safety and the $F_S$ repeatedly crosses the critical threshold $F_S = 1$ during the shock. This means that the available strength was reached since the very first oscillations and irreversible displacements grew up towards collapse. In cases similar to the Nera slide, a high $F_S$ evaluated in static conditions is not meaningful in evaluating the "distance" from failure in seismic conditions, as also for moderate shaking a very small deviation of the resultant force from the vertical direction can be sufficient to activate a less safe sliding mechanism.

Due to the geo-structural setting and the amplitude of the examined seismic shocks, the analysis of the Costa Cattiva rockslide yields a sliding mechanism along the line $i_{12}$ both in static and dynamic conditions (**Figure 8b**). The position of the blocks produced by the rock avalanche that followed the wedge failure confirms this mechanism. For both the October 26[th] shocks, the analyses do not justify the Costa Cattiva failure in seismic conditions (i.e., the computed $F_S$ is > 1). Since an overestimation of the strength is improbable due to the simple structural conditions and the low joint roughness, this result is likely explained by having neglected the small-scale amplification. The wedge was in fact located on top of a narrow sharp ridge protruding from the slope.

The analysis of Piè la Rocca slide (**Figure 8c**) also helped to assess that the wedge likely failed during the Norcia event ($M_W$=5.3). The earlier event (Accumoli, $M_W$=6.0), $F_S$ trespasses the stability threshold only once and for a very short time

span, which could cause only very small displacements without reaching full collapse. Although the displacements experienced by the rockslide and the maximum available displacement before the collapse were not estimated, the geometric layout of the rockslide scar suggests that the wedge should have experienced displacements as large as to break, at least partially, a constraining rock spur at its highest part, whose failure surface is however small (3%) compared to the area of plane 1. At Piè la Rocca rockslide, the frequent switches between the two sliding modes determine a change in $F_S$ that is not as important as for the Nera slide because the two mechanisms have quite similar safety margins against failure.

The Rubbiano rockslide (**Figure 8d**) maintains a unique mechanism during the application of the acceleration history, and despite the significant epicentral distance (7 km), the available strength was exceeded several times during the strong $M_w$-6.5 shock. The structural layout of the slide scar and the slenderness of the detached wedge (small thickness normal to the cliff in comparison to the large extent parallel to the cliff) indicate that very small displacements were sufficient to reach the collapse, likely favoured by a rocking effect.

Along the time histories shown in **Figure 8**, the values of $F_S$ are highlighted for particular instants: when the components along the geographical directions (E-W, N-S, Up-Down) and the horizontal component of the acceleration reach their maximum absolute values (respectively *x, y, z, h* points in **Figure 8**), and when the acceleration vector magnitude reaches its maximum value (*m* points in **Figure 8**). It is apparent that, despite at these instants the inertial force is quite high, $F_S$ is not always near its minimum. For some cases (e.g. *x, h, m* conditions of Costa Cattiva slide and Piè la Rocca slide during the Oct. 26th shock) these instants even correspond to the maximum values of $F_S$.

Different instantaneous $F_S$ values are also highlighted when the maximum values of particular acceleration components (all towards the slope and related to the geometry and orientation of the slope) are reached. These are the dip direction of the rock face (*n* points in **Figure 8**), the intersection line between the two main sliding planes (*i12* points in **Figure 8**), and the dip directions of the two planes (*p1* and *p2* points in **Figure 8**;). The results show that both *n* and *i12* conditions give the minimum $F_S$ or a value near to the minimum of the entire shock. The only exception is represented by the *i12* condition for the Nera slide. In this case, the resultant force falls in the region of the safer between the two possible mechanisms and *i12* condition gives $F_S = 1.5$, i.e., much higher than the minimum value ($F_S = 0.83$).

These observations provide some clues for a rational choice of the direction of the inertial force to be applied on a 3D rock wedge in pseudo-static stability analyses. Due to the significant anisotropy of the mechanical problem, the inertia calculated at instants when the magnitude of the acceleration vector or that of some pre-defined components are maximum can have negligible or favourable influence on stability. Conversely, the resistance to sliding can be overcome when the component along more adverse directions, either that along *p1/p2* or *i12*, is significant. The orientation of the resultant forces determining the minimum $F_s$ (stars in **Figure 7**) always falls within sectors delimited by these two directions (thick dashed lines in **Figure 7**). Application of the pseudo-static inertial force along these two directions yields the most conservative result only on condition that they involve all the possible sliding mechanisms. Otherwise, if both these directions correspond to the same mechanism, other orientations of the inertial force should also be tested to verify the activation of different mechanisms.

## 5 Discussion

Stability analyses in static and seismic conditions were performed on four rockslides occurred during the main shocks of the 2016-2017 Central Italy seismic sequence. The failed masses can be realistically sketched as wedges delimited by two intersecting planar discontinuities and possibly by a detachment surface at their back. The activated primary mechanisms were sliding along either one plane or the intersection line between two planes. These mechanisms developed until the wedges lost their constraints and rock falls/avalanches started. The volume of the rockslides (not exceeding 32,000 m$^3$) is small enough to assume an initial rigid motion of the wedges.

The available ground motion measurements were interpolated with an attenuation law with fixed source mechanism and stiffness class. Then a simple visco-elastic model was implemented in a parametric finite difference stress-strain analysis to calculate motion modifications due to the morphologic conditions, i.e. a step-like rock slope. Both the normalized results and the applications to the actual rockslide sites show that significant horizontal amplification is expected almost only at the crest while at intermediate heights the main effect is a reduction of the horizontal component and an amplification of the vertical one. Cyclic strength degradation is another important issue that seems to have played an important role in most of the major rockslides described in the previous sections. The high number of loading cycles applied during the main earthquakes seem to have especially affected the rock bridges along persistent joints of the limestone formations, both under shear and in tension. In this respect, static limit equilibrium back analyses of the Nera rockslide indicate that rock bridges were necessary to ensure stability even in static conditions and also provided sufficient strength to maintain the wedge stable during the October 26$^{th}$ $M_w$6.0 event. Shear strength was most likely overcome during the successive October 30$^{th}$ $M_w$6.5 shock.

The examined rockslides, which represent four of the largest landslides that occurred during the 2016-2017 sequence, are all characterized by a highly asymmetric wedge shape. This entails a low factor of safety due to the reduced (even null) strength contribution along one of the wedge planes. Even the static LEM stability analyses showed that the potential mechanisms are often not univocally established. In fact, either minor modifications of the geometrical layout or a small deviation of the resultant of external forces can activate a different mechanism with respect to that initially hypothesized. A clear representation of this problem can be obtained in the stereographic projection by subdividing the direction space into regions associated to different mechanisms. The variability of the mechanism is particularly significant in seismic conditions when the inertial force is added to the weight of the blocks thus making the resultant force fluctuating around its initial orientation.

The instantaneous LEM back analyses, carried out under the hypothesis that blocks are rigidly connected to the underlying bedrock, showed that also the safety margin can deeply fluctuate during the shock as a function of the mechanisms that are potentially activated. The minimum safety factor during the shock does not necessarily coincide with the typical directions of the pseudo-static force in a classic pseudo-static analysis (normal to the slope face or along the line

of intersection between the sliding planes). Therefore, direction is to be varied through a rational and complete examination of all the possible mechanisms.

## 6 Conclusions

From the analysis of the local seismic response and of the stability in static and seismic conditions of the four investigated rock slides, the following considerations can be highlighted:

- All the rockslides experienced significant horizontal amplification only at the crest while at intermediate heights the main effect is a reduction of the horizontal component and an amplification of the vertical one.
- Instantaneous LEM stability analyses showed that the potential mechanisms are often not univocally established, as either minor modifications of the geometrical layout or a small deviation of the resultant of external forces can activate a different mechanism and the safety margin can deeply fluctuate during the shock.
- An increased awareness on the actual failure mechanism from instantaneous stability analyses can better orient more complex analyses both for the triggering stage and the propagation phase.
- Cyclic strength degradation, due to the high number of loading cycles, likely played an important role in weakening both shear and tensile strength of the rock bridges along persistent joints of the limestone formations, at least at two sites.

The study indicates that the actual failure mechanism largely depends on the knowledge of the specific structural features of the slope. This issue is to be accounted for in risk analysis not only at local scale but also for long stretches of valley flanks overlooking transportation infrastructures in mountainous regions. In this respect, the extensive application of UAV surveys gave the possibility to obtain quantitative data even to slopes inaccessible to remote terrestrial surveys and demonstrated to be applicable at affordable times and costs along infrastructure stretches of considerable length. Quantitative data consist not only in the determination of geometry and structural setting of the slope, but also of geomechanical parameters as medium- to large-scale roughness and extent of the rock bridges along major joints.

• **Code availability** NOT APPLICABLE

• **Data availability** NOT APPLICABLE

• **Author contribution** L.V. conceptualization, formal analysis and software, writing-original, writing-review, visualization; G.F. and M.D.F.: geological investigation, writing-original, writing-review, visualization, PT: conceptualization, geotechnical investigation, funding acquisition, writing-review; GL: supervision, writing-review, funding acquisition, KF: UAV investigation, writing-review, funding; AS: investigation, supervision, funding acquisition

**Competing interests** The authors declare that they have no conflict of interest.

**Funding:**

M. De Falco, G. Forte and A. Santo were partially funded by the Italian Civil Protection Department RELUIS project (2018), PR8-UR18-WP2 UNINA (P.I. A. Santo), G. Lanzo, P. Tommasi and L. Verrucci by the projects Progetto di Ateneo Sapienza 2017 "Site investigations, monitoring and modelling of earthquake induced rockslides triggered by the 2016-2017 Central Italy seismic sequence" and Progetto di Ateneo Sapienza 2020 "Failures occurred during the 2016-2017 Central Italy seismic sequence based on site investigation and numerical modelling: hints for hazard mitigation in high seismicity areas"

(P.I. G. Lanzo), P. Tommasi by Progetto Fra.Si. funded by Italian Ministry for Ecological Transition (P.I. P. Reichenbach), K. Franke by a Mentored Research Grant (MRG) from the Brigham Young University Ira A. Fulton College of Engineering and the Center for Unmanned Aircraft Systems (C-UAS), the National Science Foundation Industry/University Cooperative Research Center (I/UCRC) under NSF Award No. IIP-1650547, along with significant contributions from C-UAS industry members.

**Acknowledgements**

Eng. Valentina Tuccio carried out preliminary seismic response analyses for the Nera River Valley rockslide. The Authority for the Sibillini Mts. National Park (Dr. A. Rossetti) and Dipartimento della Protezione Civile (Engg. P. Pagliara and P. Bertuccioli) are acknowledged for allowing surveys in the park area and for access to landslide areas during the Seismic emergency, respectively. BYU undergraduate students Bryce Berrett, Nicole Hastings, Jeffery Derricott, Bridgette Ostrum,

Doug Graff, and graduate student Michael Freeman contributed significantly to the UAV fieldwork and development of the 3D reconstructions of the rock falls. Authors thank two anonymous reviewers for their helpful suggestions.

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

**FIGURES**

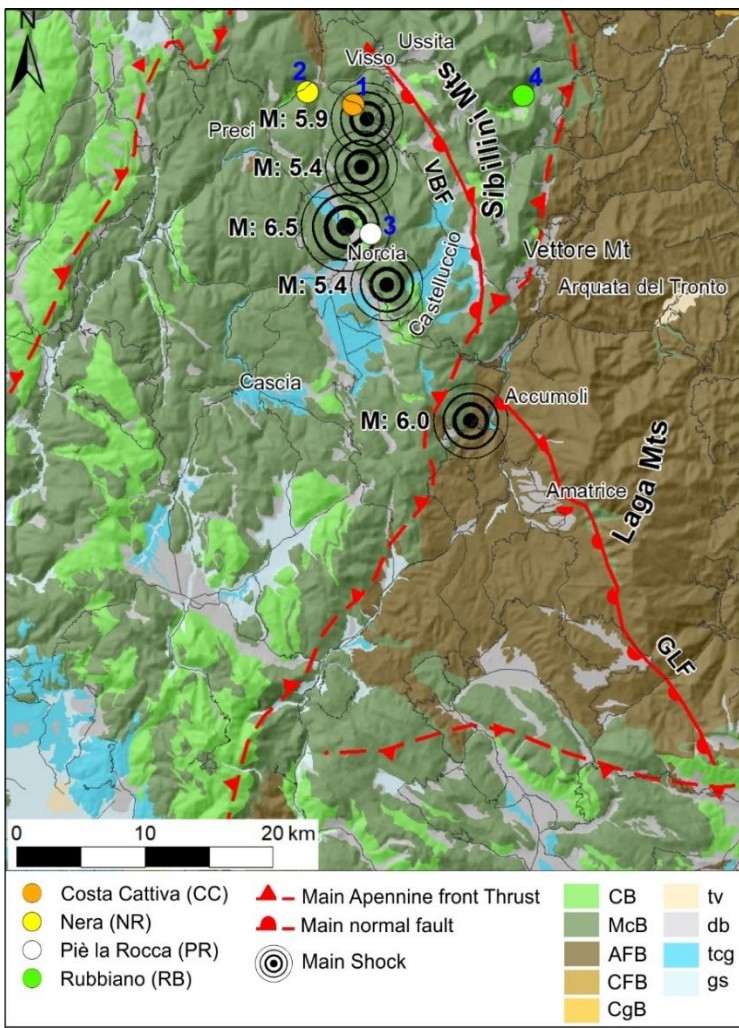


**Figure 1: Epicentres of the 2016–2017 CISS (Central Italy Seismic Sequence) and location of the studied landslides on a simplified geological map (modified after Forte *et al.*, 2019). Keys: CB Carbonate Bedrock; McB Marly Carbonate Bedrock; AFB Arenaceous Flysch Bedrock; CFB Clayey Flysch Bedrock; CgB Conglomerate Bedrock; tv Travertine; db Debris; tcg terraced conglomerates; gs gravels and sands. GLF Gorzano – Laga Fault; VBF Vettore – Bove Fault.**

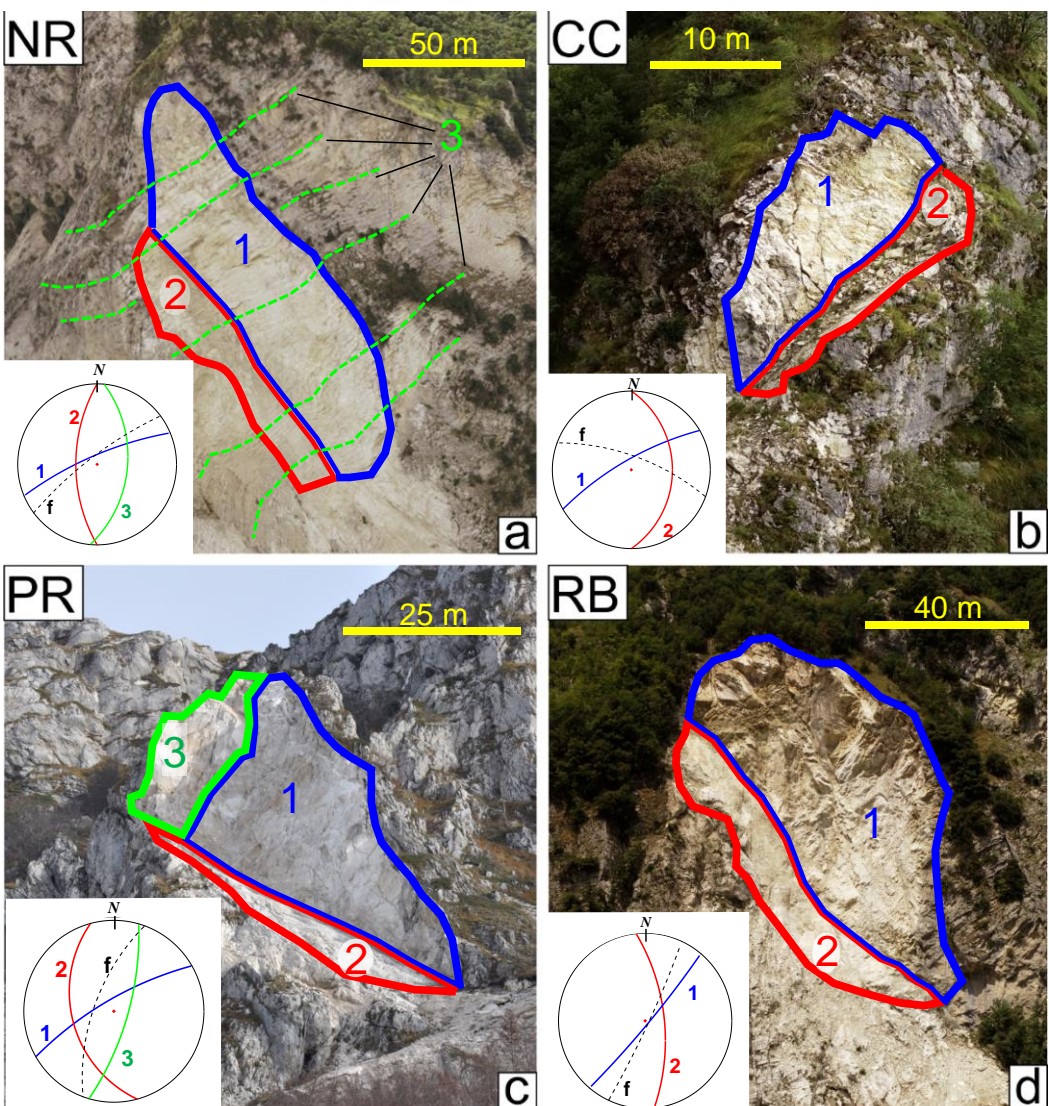

**Figure 2: Frontal view of the scars of the studied rockslides with approximated limits of the mean delimiting planes; a) Nera (NR); b) Costa Cattiva (CC); c) Piè la Rocca (PR); d) Rubbiano (RB). In the inner boxes: stereographic projections (lower hemisphere) of the discontinuity planes delimiting the failed masses (1, 2, 3) and of the local slope face (f). Details in Forte et al. (2021).**



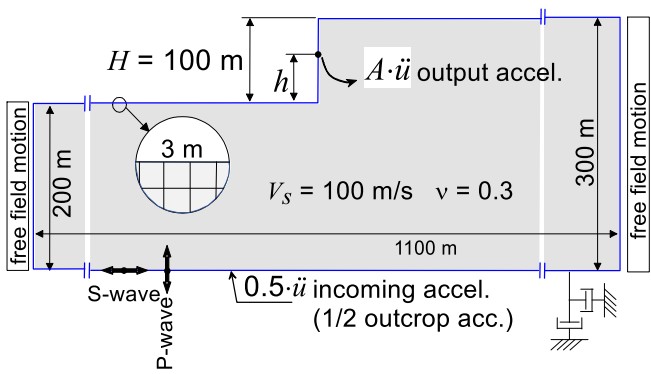

**Figure 3: Numerical model used for the seismic response of a vertical cliff.**

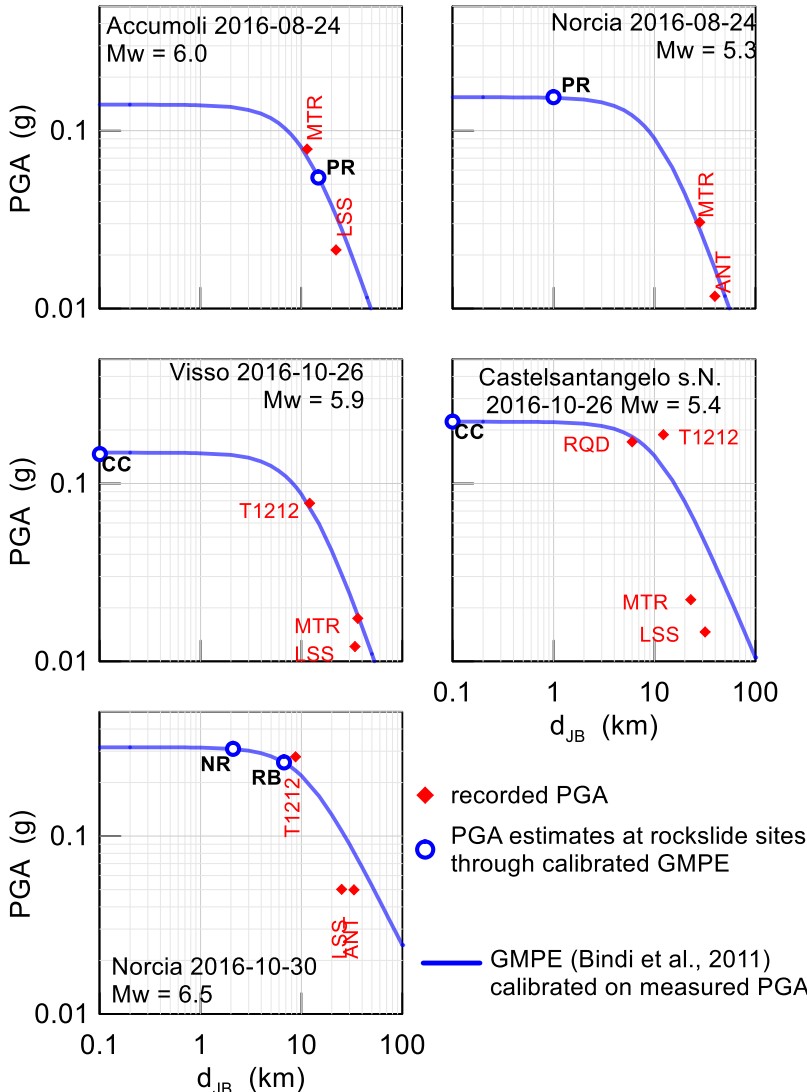


**Figure 4: PGA estimate through a calibrated GMPE for each shock. For PR rockslide and CC rockslide, two shocks are considered on August 24th and on October 26th, respectively.**

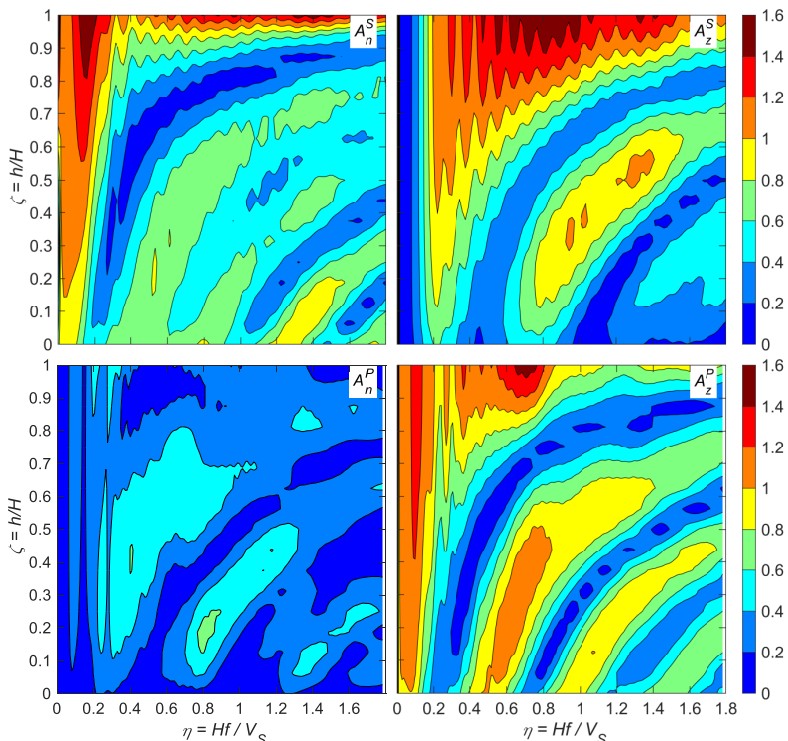

**Figure 5: Amplification ratios with respect to the outcrop motion along the vertical wall of a step-like slope. Incident S wave ($A_n{}^S$, $A_z{}^S$) and incident P wave ($A_n{}^P$, $A_z{}^P$) (b).**

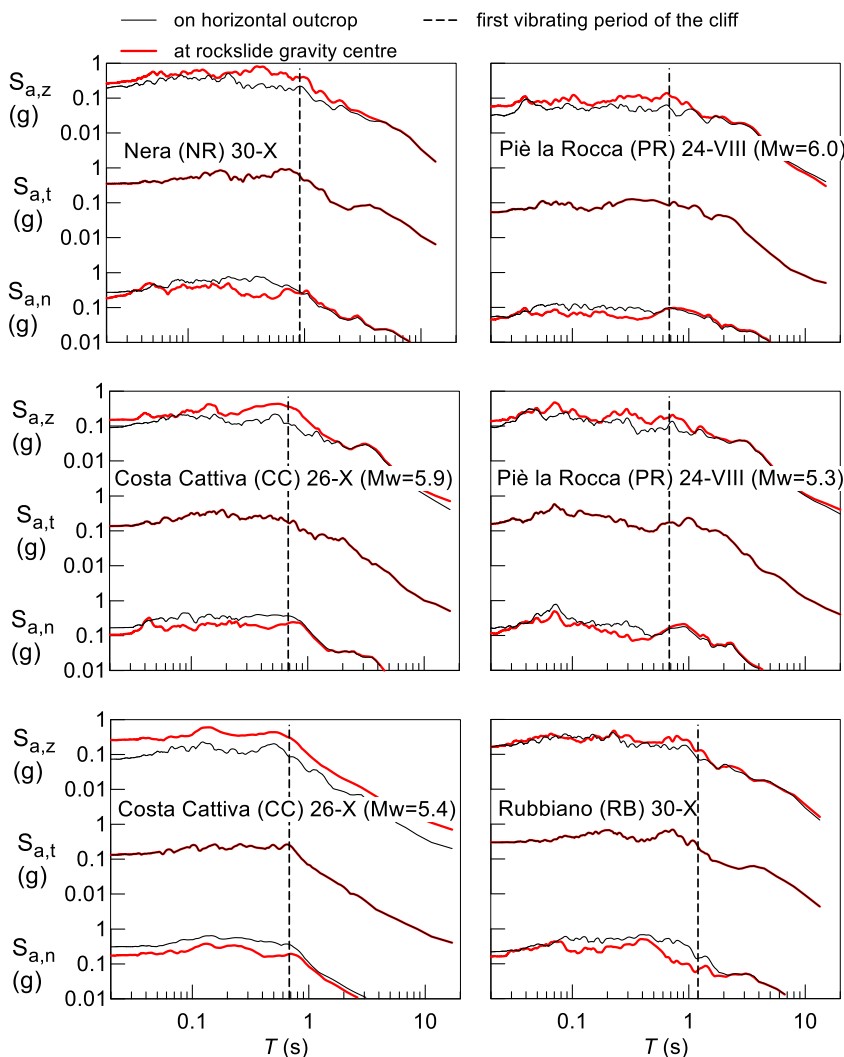

**Figure: 6: Response spectra of the three components of the acceleration at the rigid horizontal outcrop and on the vertical rock cliff at the elevation of the rockslide centre of mass as estimated through the seismic response of the numerical model for the four case studies (for each possible triggering earthquake).**




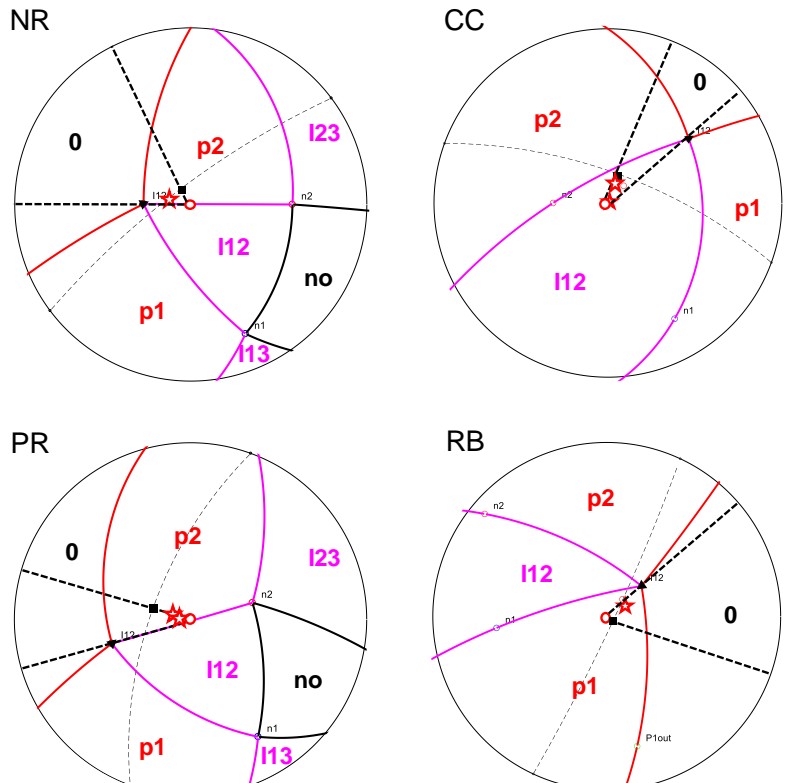

**Figure 7: Stereographic conform projections (lower hemisphere projected from upper focal point) of the trihedral/dihedral regions (solid lines) that define different sliding mechanisms depending on the direction of the resultant force.** $I_{ij}$=sliding along the intersection line between the planes *i* and *j*, *pi*=sliding along the plane *i*. **Light dashed lines are the projections of the average local slope face. Red stars indicate the resultant orientations corresponding to the minimum** $F_s$ **during the seismic shocks (see Fig. 8). Full triangles and squares indicate orientation of the intersection line ($I_{12}$) between the two planes and the dip direction of the slope face, respectively.**


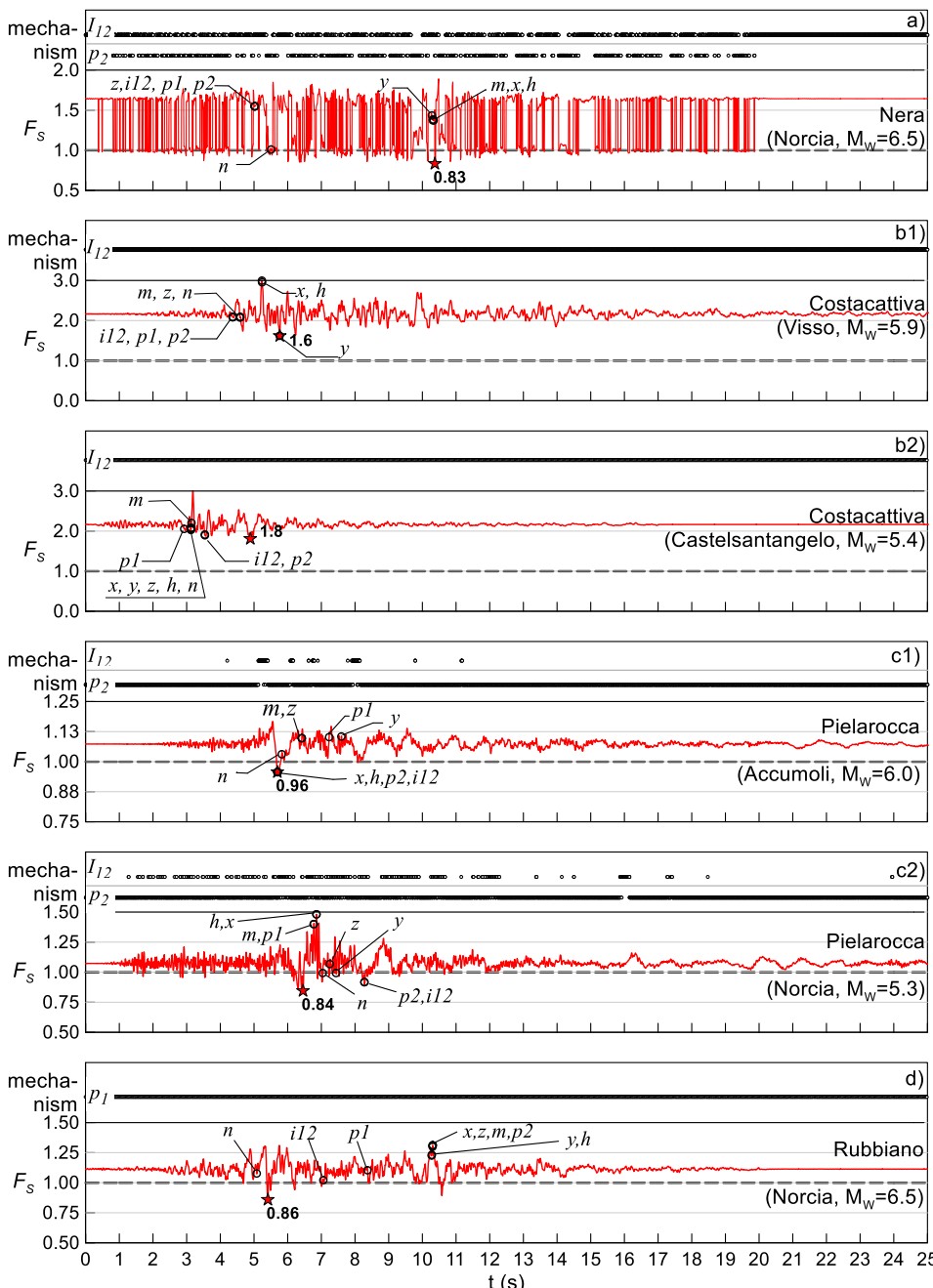

**Fig 8 Time histories of the safety factor $F_s$ during the triggering shocks and of the instantaneous active mechanism: $I_{12}$ = sliding along the intersection line between the planes, $p_1$, $p_2$ =sliding along the plane 1 and 2 respectively. Stars indicate minimum $F_s$. Empty circles highlight instants with peak values of specific acceleration components: $x,y$ = geographic components (E,N), $z$ = vertical component, $h$ = horizontal component, $n$ = component normal to the slope face, $i12$ = intersection line of the sliding planes, $p1$ and $p2$ = dip directions of the sliding planes, $m$ = instant of maximum acceleration magnitude.**

**Table 1: Main features of the investigated rockslides**

| Landslide | | Possible triggering Earthquakes (date, GMT, moment magnitude) | Estimated volume (m³ x 10³) | Lithology |
|---|---|---|---|---|
| Nera (Sasso Pizzuto Mt.) | NR | October 30th 2016, 6:40:18, $M_W$=6.5 | 32.0 | *Layered limestones (Maiolica Fm.)* |
| Costa Cattiva (Nera River Valley) | CC | October 26h 2016, 17:10:36, $M_W$=5.4 October 26h 2016, 19:18:06, $M_W$=5.9 | 0.4 | " |
| Rubbiano (Infernaccio gorge) | RB | October 30th 2016, 6:40:18, $M_W$=6.5 | 15.0 | " |
| Piè la Rocca (Patino Mt.) | PR | August 24th 2016, 01:36:32, $M_W$=6.0 August 24th 2016, 02:33:29, $M_W$=5.3 | 15.0 | *Massive limestones (Calcare Massiccio Fm.)* |

**Table 2: Peak ground accelerations from the available records of the shocks at stations installed on rigid outcrop within 50 km**
**from the epicentres of the seismic events.**

| Event | | | Seismic station | Epicentral distance | $D_{JB}$ | Horizontal E-W PGA | Horizontal N-S PGA | Vertical PGA |
|---|---|---|---|---|---|---|---|---|
| Epicentre | Date | Mw | | km | km | m/s² | m/s² | m/s² |
| Accumoli | 2016-08-24 | 6.0 | MTR | 19.40 | 11.40 | 0.791 | 0.754 | 0.418 |
| | | | LSS | 26.70 | 22.22 | 0.230 | 0.190 | 0.151 |
| Norcia | 2016-08-24 | 5.3 | MTR | 30.80 | 28.13 | 0.295 | 0.305 | 0.137 |
| | | | ANT | 42.00 | 39.68 | 0.118 | 0.112 | 0.048 |
| Visso | 2016-10-26 | 5.9 | T1212 | 18.8 | 12.1 | 0.667 | 0.866 | 0.445 |
| | | | LSS | 41.1 | 33.9 | 0.128 | 0.110 | 0.118 |
| | | | MTR | 43.8 | 36.2 | 0.174 | 0.168 | 0.091 |
| Castelsantangelo sul Nera. | 2016-10-26 | 5.4 | T1212 | 15.2 | 12.3 | 1.767 | 1.917 | 0.588 |
| | | | RQT | 17.4 | 3.7 | 2.177 | 1.296 | 0.880 |
| | | | LSS | 37.6 | 23.8 | 0.148 | 0.139 | 0.079 |
| | | | MTR | 40.9 | 22.6 | 0.179 | 0.266 | 0.085 |
| Norcia | 2016-10-30 | 6.5 | T1212 | 10.50 | 8.77 | 2.744 | 2.731 | 1.636 |
| | | | LSS | 32.60 | 25.10 | 0.464 | 0.523 | 0.399 |
| | | | ANT | 46.10 | 33.27 | 0.436 | 0.546 | 0.242 |

**Table 3. Input parameters for the static LEM stability analyses after Forte *et al.* (2021)**

| rockslide | volume m³ | plane 1 | | | plane 2 | | | plane 3 | LEM (static condition) | |
|---|---|---|---|---|---|---|---|---|---|---|
| | | dip/dd °/° | $\varphi_1'$ ° | $A_{rb1}$ m² | dip/dd °/° | $\varphi_2'$ ° | $A_{rb2}$ m² | dip/dd °/° | mechanism | $F_S$ - |
| NR | 30940 | 77/337 | 47 | 570+800 | 60/270 | 40 | 0 | 48/95 | line $I_{12}$ | 1.68* |
| CC | 400 | 75/330 | 47 | 0 | 35/090 | 40 | 0 | - | line $I_{12}$ | 2.16 |
| PR | 14000 | 75/330 | 47 | 0 | 40/255 | 42 | 0 | 72/106 | on plane 2 | 1.07 |
| RB | 15000 | 65/084 | 47 | 0 | 85/130 | 40 | 2880 | - | on plane 1 | 1.11** |

\*: with the cohesive contribution of the spur at the lower wedge tip (800 m²)

\*\*: with the tensile contribution of the rear wedge surface (composite surface labelled as plane 2)

$A_{rb}$: areas of intact rock along the sliding planes providing cohesive contribution

**Table 4: Parameters of the triggering events utilized to calculate the motion at the rockslide sites from the available recorded accelerograms**

| Rockslide | | | Seismic event | | | Epicentral distance | $D_{JB}$ | $S$ | PGA [*] |
|---|---|---|---|---|---|---|---|---|---|
| Site | lat. | long. | | | | | | | |
| | ° | ° | epicentre | date | $M_W$ | km | km | - | g |
| NR | 42.93 | 13.07 | Norcia | 2016-10-30 | 6.5 | 10.2 | 2.1 | 1.103 | 0.309 |
| RB | 42.93 | 13.28 | Norcia | 2016-10-30 | 6.5 | 16.8 | 6.7 | 0.926 | 0.259 |
| CC | 42.92 | 13.12 | Visso | 2016-10-26 | 5.9 | 2.3 | 0.0 | 1.862 | 0.146 |
| | | | Castelsant. | 2016-10-26 | 5.4 | 4.4 | 0.0 | 1.186 | 0.223 |
| PR | 42.82 | 13.13 | Accumoli | 2016-08-24 | 6.0 | 4.6 | 1.0 | 0.668 | 0.054 |
| | | | Norcia | 2016-08-24 | 5.3 | 16.3 | 14.8 | 1.890 | 0.154 |

[*] peak ground acceleration estimated at the site on rigid horizontal outcrop

**Table 5: Parameters utilized to calculate the topographic modification of the seismic motion at the rockslide sites**

| Rockslide | Dip direction of the slope face, $\alpha$ | Cliff height, $H$ | Height of the rockslide centre of mass, $h$ | Period of first mode, $T_0$ |
|---|---|---|---|---|
| | (°) | (m) | (m) | (s) |
| Nera (NR) | 330 | 400 | 250 | 0.91 |
| Costa Cattiva (CC) | 330 | 300 | 90 | 0.68 |
| Rubbiano (RB) | 115 | 530 | 170 | 1.20 |
| Piè la Rocca (PR) | 330 | 300 | 180 | 0.68 |