# Peer review of "Instantaneous LEM back-analyses of major rockslides triggered during the 2016-2017 Central Italy seismic sequence"

_Natural Hazards and Earth System Sciences, 2022_

## Author Response (AR1)

**REVIEWER 1**

The manuscript presents an interesting approach for analyzing earthquake-induced wedge failures, modeling the evolution of the factor of safety during earthquakes using instantaneous pseudo-static analyses, while taking into account real seismic records and topographic amplification for the ground motion inputs. The methodology and results are a contribution to the knowledge of coseismic landslides. The paper relies too much on previous publications from some of the authors, such as it lacks the necessary context for the reader. Some of the assumptions made for the analyses and some figures need some further justification or explanation (see below). These changes can be achieved with a moderate revision.

Specific comments:

R Lines 59 to 62: This paragraph is insufficient. A summary of the geology, seismotectonic setting including the faults where the earthquakes originated, and a description of the 2016 earthquake sequence are required. The reader needs some context to understand the analyses without referring to other publications.

A We agree with the reviewer and the text was modified accordingly.

R Line 74: Check redaction. "Ground modification induced by stratigraphic conditions were not considered because all slides are in bedrock...."?

A The text was modified

R Line 90 and Table 3: explain the scaling procedure to obtain S

A The text was modified defining S=PGA\*/PGArec where PGA\* is the horizontal PGA estimated at the site through the interpolated GMPE curve, while PGArec is the peak horizontal component recorded at the closest seismic station on rock.

R Lines 98 to 105: How steep are the slopes? You may add that slope gradient data in Table 3, or some cross sections of the landslides, to justify that the vertical cliff model is a reasonable approximation for all the analyzed landslides.

A The mean dip of the slope was added through a new column in Table 4

R Line 108, Figure 5 caption and elsewhere: you use the term "horizontal rigid outcrop" to refer to site of the reference ground motion for topographic amplification calculation. I presume this is at the base (the top is also horizontal), and at some distance from the cliff (it can be attenuation at the slope toe). Please clarify this location and maybe use a different name for it.

A The test was modified defining the reference motion as "the horizontal rigid outcrop at lower elevation in free field conditions"

R Line 139: check the phrase "the inverse Fourier transform to the (1) and....", should be "to equation (1)"?

A the text was modified.

R Lines 144 and 145, Figure 5. You say that "The alteration of the motion is usually significant for periods lower than 1-2 s, while it is negligible for periods higher than the fundamental period T0...". However, in Fig. 5 it does not look negligible between To and 2 seconds in some of the charts, which agrees your first phrase. Check the description and correct the second part if needed.

A The text was modified

R Lines 174 to 182. Please comment on the validity of factors of safety below 1.0 after the first time this value is reached and some sliding occur.

A The comment was added.

R Lines 196 to 201: Please add which software or code did you use to make the factor of safety calculations.

A The comment was added.

R Lines 201 to 203: The last sentence of this paragraph could go in the Discussion section

A The last sentence was moved to the Discussion section

R Lines 215 and 216: "the geometric layout of the rockslide scar suggests that the wedge should have experienced displacements as large as to break a constraining rock spur at its highest part" That's vague, how large is that? an estimate at least? What size is the "spur"?

A The text was modified

R Lines 237 to 246: The whole paragraph should better go in the Discussion section.

A The paragraph was moved to the Discussion section

R Figure 3: Please enlarge the size of blue dots, they are hardly distinguished.

A Done

R Figure 7 caption: Indicate what is the meaning of the black lines at the top of each chart (mechanism).

A The figure and the caption were modified

R Table 2: I suggest adding the distance between the seismic station and the landslide of interest for which you use the ground motion records.

A The data about the distance of the seismic station was added in Table 3

R Table 2: Better use units of "g" for accelerations, to be consistent with Table 3.

A Done

R Table 3: add average slope gradient at each rockslide site

A The mean slope dip was added in table 4

R Table 5: Add the cohesion values, even if it is the same for all.

A Done

**REVIEWER 2**

In this paper, the authors examine four wedge failures triggered by earthquakes that occurred in Central Italy. The authors use 2D mechanical models to better understand the stability of the examined hillslopes under the influence of seismic forces. I believe the manuscript could be significantly improved by reviving the general structure of the manuscript. In the current version of the manuscript, data, method and results are presented together and that makes it a bit difficult to follow sometime. The research gap(s) and research question could be better emphasized. Also, the relevant literature could be enriched.

Below I've included line by line suggestions and highlighted all these points.

- R Line 13. "occurred" This is a statement based on reported events so, it is better saying "Most of the reported/documented landslides"
- A Four of these failures, including the three largest among the documented landslides, were described in terms of structural and geomechanical investigations in a previous study.
- R Also at Line 35
- A These considerations sparked investigation of the failure stages of the largest rockslides among those reported during the 2016-2017 Central Italy seismic sequence (CISS)
- R Line 23. "Lombardo et al. 2021, Quinton Aguilera et al. 2022" I do not think these are papers reporting rockslide volumes. Could you please replace them with relevant articles?
- A Citation (Lombardo et al. 2021, Quinton Aguilera et al. 2022) has been replaced with:

Malamud et al. 2004; Marc et al. 2017

- R Lines 23-24. "for the energy released by these seismic events (moment magnitude, Mw< 6.5)" Could you please revise this line, I could not get what you mean here.
- A [...] due to the energy released by these seismic events (moment magnitude, Mw < 6.5).

- R Lines 25-27. You are referring to co-seismic rockslides reported in Central Italy, right? Please indicate that here again for clarification. Also, it would be better if you give citations to the literature. What is the source of these descriptions?
- A Similarly to many other earthquake-triggered landslides (Rodriguez et al. 1999), landslide triggered by Central Italy earthquakes were characterized by a marked disruption of the rock mass and originated on steep slopes, where inertial forces easily remove welldelimited and scarcely constrained blocks from the slope through rigid sliding/toppling or tensile failures of overhanging blocks (Esposito et al. 2000, Lanzo et al. 2009, Franke et al. 2019)

Lanzo G., Di Capua G., Kayen R.E., Kieffer D.S., Button E., Biscontin G., Scasserra G., Tommasi P., Pagliaroli A., Silvestri F., d'Onofrio A., Violante C., Simonelli A.L., Puglia R., Mylonakis G., Athanasopoulos G., Vlahakis V., Stewart J.P. (2009). Seismological and geotechnical aspects of the Mw=6.3 L'Aquila earthquake in central Italy on 6 April 2009. International Journal of Geoengineering Case histories, http://casehistories.geoengineer.org,1(4):206-339.

- R Line 31. "local" For strong earthquakes, it could be also important for regional scale assessments, no?
- A Since seismic loading acts only at the early detachment of earthquake-triggered rock failures (propagation is controlled only by gravity loading and slope geometry) the study of this stage is very important for local hazard evaluation

R Lines 30-35. Yes, you can say that this is an important concept for co-seismic landslide hazard assessment. I believe this is the point where you should emphasize the gap in the literature that you are aiming to address with this research. After emphasizing the research gap, instead of saying "These considerations sparkled investigation of the failure stages of the largest rockslides ...", please be more specific and indicate the specific research question that you are targeting in this paper. And then, you can mention what you did and how you did that goal you described. Just a small suggestion to improve the flow of the manuscript and to clarify your point.

A The text was accordingly modified.

- R Lines 41-57. Please trim this part because part of the content presented here would be better if you present it in the method section.
- A This part has been trimmed as suggested.
- R Line 58. I recommend using the traditional structure as Intro, study area/materials/data, method, discussion and conclusions.
- A The structure has been significantly modified through adding a new (3rd) section about the method and moving all data description into the 2nd section
- R Lines 59-60. I agree with the authors that detail information regarding geology, rock mass structure and so on may not need to be presented here if there is already a paper describing the very same. However, you can still briefly introduce your study area briefly. Please start by introducing your study area then after your brief summary you can mention Forte and others for further details.
- A A brief introduction was provided.
- R And I believe you should tell us why you chose those four landslides in particular. Do you have detailed geotechnical information about them? Please elaborate.
- A These rockslides were chosen because they represent four of the largest failures among those detected during the reconnaissance field surveys conducted immediately after the seismic shocks (Costa Cattiva and Nera rockslides) or the most accessible among those observed on aerial images taken soon after the end of the seismic sequence (Piè la Rocca and Rubbiano rockslides). In this way, UAV surveys, which allowed detailed morphological and geo-structural setting, could be conducted in a relatively short time after the Seismic sequence. Other large rockslides detected on aerial images, with much higher logistic problems, were successively investigated and are currently being analyzed in the framework of national research projects.

The manuscript was modified in this way:

..... Figures 2a through 2d present post-collapse frontal views of the rockslides. All the failure scars are carved in sound limestone. either layered (Costa Cattiva and Nera rockslides) or relatively massive (Piè la Rocca and Rubbiano rockslides). The four rock slopes are all very steep and three of them (Nera, Piè la Rocca and Rubbiano) are located within tectonically disturbed zones: (reverse fault and associated fold hinge, a fault zone, and a thrust front, respectively). The wedges were all delimited by single near-planar major joint surfaces (labelled in Figure 2), excepting for the Rubbiano rockslide, which was delimited at its back by a combination of several discontinuities of fairly limited extent. Figure 2 also includes stereo-plots with great circles of the planes delimiting each wedge at the very beginning of the detachment, as estimated from the 3D models and point clouds obtained from UAV aerial surveys (Franke et al. 2019; Tommasi et al. 2019). Great circles refer to single major joints or to planes interpolating combinations of minor joints. Low-dip joints (i.e., along which shear occurred) showed negligible intact rock bridges excepting for that delimiting at the base the Nera rockslide and its contribution to shear strength was therefore considered. Portions of intact rock were found along the subvertical surfaces delimiting the back of two of the failed wedges, where they provided some tensile resistance: the Piè la Rocca and the Rubbiano rockslides. The latter was larger enough to deserve consideration in the stability analyses.

R Line 66. Figure 2 is not something you adapted from the literature, right? If this is something you generated as part of this research, please keep it for your result section because this is also part of your results; you carried out some kinematic analyses and generated those plots. First, mention in the method section and then present in your results section.

Data and results of Fig.2 are derived from the research described in Forte et al. (2021). The citation has been added in the caption.

- R Line 70. This should be your method section now. Of course, you can add sub-headings.
- A See the response about the new text structure.

- R Lines 76-82. This paragraph could be moved to the previous section (i.e., study area/materials/data) as you are presenting your landslides here.
- A The paragraph has been moved into the 2nd section (data).

R Line 87. Remove "."

A Done

- R Lines 104-105. Is this an important literature gap that you aim to address? If this is the case, please mention it in your introduction.
- A The suggested modification has been applied
- R Lines 110-112. Could you please cite the relevant literature? You can provide more details through Supplementary Materials maybe. Specifically, topographic amplification deserves a better summary. It should be also better elaborate in the introduction. The given literature on this topic should be improved.
- A The topographic amplification has been inserted into the introduction. The literature on the topographic amplification was limited to the cases of rock slopes because the general topic of the topographic amplification is quite wide, and it should be not suited for this paper.
- R Line 122. Assimaki et al. (2005) are focusing on soil, right? In this regard, can we assume that the amplification mechanism will be the same or similar?
- A The numerical analysis of Assimaki et al. (2005) whose results are compared to the results of the present research are the preliminary dynamic analyses for elastic slope described section. Therefore the non-linear behaviour is not considered for these results.
- R Lines 123-130. These are part of your results, aren't they?
- A This part has been collected with the other results.

R Line 139. Could you please elaborate on these assumptions? Based on what? The same comment is valid for the whole section.

- A The assumption has been better explained: "The hypothesis that the seismic response develops in plane strain condition can be assumed for very long cliff or valley and therefore the component ap(t) can be considered unmodified"
- R Section 4. Please consider the very same comment I mentioned above. The parts that you present your method or data could be differentiated from your results and also assumptions and knowledge you obtained from the literature could be better explained.
- A Section 3 and 4 in the new structure has been dedicated only to the new results (see the response about the text structure).